# Transferability of features for neural networks links to adversarial attacks and defences

**Shashank Kotyan** [1]*, **Moe Matsuki**[2], **Danilo Vasconcellos Vargas**[1,3]

**1** Department of Information Science and Engineering, Kyushu University, Fukuoka, Japan, **2** SoftBank Group Corporation, Tokyo, Japan, **3** Department of Electrical Engineering and Information Systems, School of Engineering, The University of Tokyo, Tokyo, Japan

* kotyan.shashank.651@s.kyushu-u.ac.jp

## Abstract

The reason for the existence of adversarial samples is still barely understood. Here, we explore the transferability of learned features to Out-of-Distribution (OoD) classes. We do this by assessing neural networks' capability to encode the existing features, revealing an intriguing connection with adversarial attacks and defences. The principal idea is that, "if an algorithm learns rich features, such features should represent Out-of-Distribution classes as a combination of previously learned In-Distribution (ID) classes". This is because OoD classes usually share several regular features with ID classes, given that the features learned are general enough. We further introduce two metrics to assess the transferred features representing OoD classes. One is based on inter-cluster validation techniques, while the other captures the influence of a class over learned features. Experiments suggest that several adversarial defences decrease the attack accuracy of some attacks and improve the transferability-of-features as measured by our metrics. Experiments also reveal a relationship between the proposed metrics and adversarial attacks (a high Pearson correlation coefficient and low p-value). Further, statistical tests suggest that several adversarial defences, in general, significantly improve transferability. Our tests suggests that models having a higher transferability-of-features have generally higher robustness against adversarial attacks. Thus, the experiments suggest that the objectives of adversarial machine learning might be much closer to domain transfer learning, as previously thought.

## 1 Introduction

Adversarial samples are noise-perturbed samples that can fail neural networks for tasks like image classification. Since they were discovered by [1] some years ago, both the quality and variety of adversarial samples have grown. These adversarial samples can be generated by a specific class of algorithms known as adversarial attacks [2–5].

Most of these adversarial attacks can also be transformed into real-world attacks [6–8], which confer a big issue as well as a security risk for current neural networks' applications. Despite the existence of many variants of defences to these adversarial attacks [9–20], 'no known learning algorithm or procedure can defend consistently' [21–26]. This shows that a

**Data Availability Statement:** A description of the data set and the third-party source CIFAR-10: The CIFAR-10 dataset consists of 60000 32x32 colour images in 10 classes, with 6000 images per class. There are 50000 training images and 10000 test

images. Homepage: https://www.cs.toronto.edu/~kriz/cifar.html Article Link: https://www.cs.toronto.edu/~kriz/learning-features-2009-TR.pdf Fashion-MNIST: Fashion-MNIST is a dataset of Zalando's article images consisting of a training set of 60,000 examples and a test set of 10,000 examples. Each example is a 28x28 grayscale image, associated with a label from 10 classes. Homepage: https://github.com/zalandoresearch/fashion-mnist arXiv Paper Link: http://arxiv.org/abs/1708.07747 ImageNet (ILSVRC 2012): ILSVRC 2012, commonly known as 'ImageNet' is an image dataset organized according to the WordNet hierarchy. Each meaningful concept in WordNet, possibly described by multiple words or word phrases, is called a "synonym set" or "synset". There are more than 100,000 synsets in WordNet, majority of them are nouns (80,000+). In ImageNet, we aim to provide on average 1000 images to illustrate each synset. Images of each concept are quality-controlled and human-annotated. In its completion, we hope ImageNet will offer tens of millions of cleanly sorted images for most of the concepts in the WordNet hierarchy. It consists of 1281167 colour training images and 50000 colour validation images covering 1000 classes. Homepage: http://image-net.org/ DOI:10.1007/s11263-015-0816-y All necessary contact information others would need to apply to gain access to the data CIFAR-10 and Fashion MNIST datasets can be downloaded directly either from the homepages of the datasets. ImageNet dataset requires registration on http://www.image-net.org/download-images in order to get the link to download the dataset. a) Please clarify in detail whether interested researchers can replicate your study findings in their entirety by directly obtaining the data from the third party and following the protocol in your Methods section. Any interested researchers can replicate our study findings entirely by obtaining the data from the dataset's website and using our publicly released code or by following the protocol which is mentioned in our Methods section.

**Funding:** DV, JST, ACT-I Grant Number JP-50243, https://www.jst.go.jp/kisoken/act-i/index.html DV, JSPS KAKENHI Grant Number JP20241216, https://www.jsps.go.jp/english/e-grants/.

**Competing interests:** The authors have declared that no competing interests exist.

more profound understanding of the adversarial algorithms is needed to formulate consistent and robust defences.

Several works have focused on understanding the reasoning behind such a lack of robust performance. It is hypothesised in [9] that neural networks' linearity is one of the main reasons for failure. Other investigation by [27] shows that with deep learning, neural networks learn false structures that are simpler to learn rather than the ones expected.

Moreover, researches by [28, 29] unveil that adversarial attacks are altering where the algorithm is paying attention. In [30], it is discussed that an adversarial sample may have a different interpretation of learned features than the benign sample. The authors show that learned features of adversarial samples are remarkably similar to different images of different true-class and links adversarial robustness to features learned by deep neural networks.

## 1.1 Overview

This article tries to open up a new perspective on understanding adversarial algorithms based on evaluating the transferability-of-features of In-Distribution classes to Out-of-Distribution classes. We do this by verifying that this transferability is indeed linked with the adversarial attacks and defences for neural networks. Specifically, we propose a methodology loosely based on Zero-Shot Learning entitled Raw Zero-Shot for evaluating this transferability.

We conduct experiments over the soft-labels of an Out-of-Distribution class to assess the transferability-of-features for various classifiers. This is based on the hypothesis that, if a classifier is capable of learning useful features, an Out-of-Distribution class would also be associated with some of these features learned from In-Distribution classes (Amalgam Proportion) (Fig 1).

We call this type of inspection over Out-of-Distribution class, Raw Zero-Shot. (Section 3) Furthermore, we also introduce two associated metrics to evaluate this transferability. One is based upon the Clustering Hypothesis (Section 3.1), while the other is based on Amalgam Hypothesis (Section 3.2).

## 1.2 Contributions

- Evaluate a wide assortment of datasets and classifiers and assess their transferability-of-features. (Section 5)

- Evaluate different adversarial defences and understand their effect on transferability-of-features. Also, determine the statistical relevance of defences on transferability by conducting a paired samples t-test. (Section 6)

- Reveal an intriguing connection between the transferability-of-features and attack susceptibility by calculating the Pearson coefficient of proposed metric measuring transferability-of-features with adversarial attacks. (Section 7)

- Discuss some observations for different adversarial attacks, defences and architectures from the perspective of transferability-of-features (Section 5)

## 2 Related works

### 2.1 Understanding adversarial attacks

Since the discovery of adversarial samples by [1], many researchers have tried to understand the adversarial attacks. It is hypothesised in [9] that neural networks' linearity is one of the

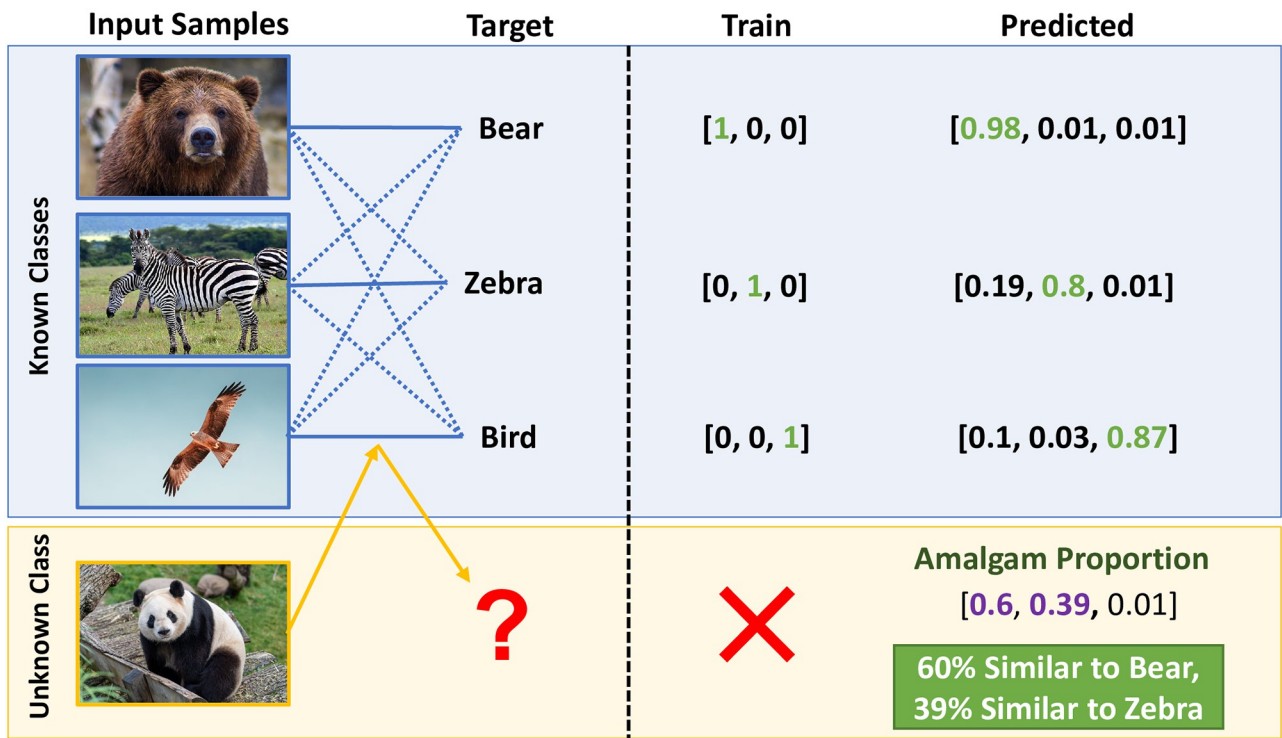

**Fig 1. Illustration of Raw-Zero Shot methodology using transferability-of-features.** In the figure, the unknown class (Giant Panda) is represented as a combination of known classes (Bear, Bird, and Zebra).

principal reasons for failure against an adversary and non-linear neural networks are thus, more robust compared to linear networks [31]. A geometric perspective is analysed in [32], where it is shown that adversarial samples lie in shared subspace, along which the decision boundary of a classifier is positively curved. In [33], a relationship between sensitivity to additive perturbations of the inputs and the curvature of the decision boundary of deep networks is shown. Another aspect of robustness is discussed in [18], where authors suggest that the capacity of the neural networks' architecture is relevant to the robustness. It is also stated in [34] that the adversarial vulnerability is a significant consequence of the dominant supervised learning paradigm and a classifier's sensitivity to well-generalising features in the known input distribution. In [35], the authors proposed a decision based black-box adversarial attacks, further suggesting that the decision boundary of neural networks might not be robust enough. Also, research by [36] argues that adversarial attacks are entangled with the interpretability of neural networks as results on adversarial samples can hardly be explained. The bounds for the robustness using this input feature space is also studied in [37]. Further, the existence of different internal representations learned by neural networks for an adversarial sample compared to a benign sample is shown in [30]. In this article, we explore a new perspective to understand adversarial attacks and defences based on the transferability-of-features of the neural networks.

## 2.2 Zero-Shot learning

Zero-Shot learning is a method to estimate Out-of-Distribution classes that do not appear in the training data. The motivation of Zero-Shot learning is to transfer knowledge from In-

Distribution classes to Out-of-Distribution classes. Existing methods address the problem by estimating Out-of-Distribution classes from an attribute vector defined manually for all classes. For each class, whether such an attribute (like colour, shape) relates to the class or not is represented by one or zero. [38] introduced *Direct Attribute Prediction (DAP)* model, which learns each parameter of the input sample for estimating the attributes of the sample from the feature vector generated. It estimates an unknown class of the source data which is estimated from the target data by using these parameters. This approach projects feature vectors generated by learned classes into the source domain to classify the unknown classes. Based on this research, other zero-shot learning methods have been proposed, which uses an embedded representation generated using a natural language processing algorithm instead of a manually created attribute vector [39–43]. The opposite direction was proposed in [44], which learned how to project from the source domain to the generated feature vector. [45] proposed a different strategy by constructing the histogram of known classes distribution for an unknown class to estimate unknown classes. [46] suggested that generative models learn better about the Out-of-Distribution class by learning independednt attributes in zero-shot setting. It is also shown in [47], that generative models are favorable for zero-shot learning. They assume that the unknown classes are the same if these histograms generated in the prediction and source domains are similar. Our Raw Zero-Shot test is distinguished from other zero-shot learning algorithms as in Raw Zero-Shot, the neural network has no access to features (attribute vector) or additional supplementary knowledge.

## 3 Raw Zero-Shot

Raw Zero-Shot is a learning test in which only $N - 1$ of the $N$ classes in the dataset are presented to the classifier during training; in other words, all the samples of one specific class are removed from the standard training dataset. Such a classifier trained on only $N - 1$ of the $N$ classes is called *'Raw Zero-Shot Classifier'*. Please note that a *'Standard Classifier'* is trained on all $N$ classes has $N$ soft-label dimensions in the soft-label space. In contrast, a Raw Zero-Shot Classifier has only $N - 1$ soft-label dimensions in the soft-label space due to the forced exclusion of a class.

The excluded (now) Out-of-Distribution class then can be predicted as a combination of the remaining $N - 1$ soft-label dimensions of the learned In-Distribution classes. We call this combination as *'Amalgam Proportion'* (Fig 1). Only the Out-of-Distribution class (excluded class from $N$) is provided to the classifier during testing. Amalgam Proportion for the given Out-of-Distribution class is recorded for the classifier. This process is iterated for all potential ($N$) classes, excluding a different class each time.

Soft labels of a classifier composes a space in which a given image would be categorised as a weighted vector involving the previously learned classes. If neural networks can learn the features existing in the In-Distribution classes, it would be reasonable to consider that the Amalgam Proportion also describes a given image as a combination of the previously learned classes (Fig 1). Similar to a vector space in linear algebra, the soft-labels can be combined to describe Out-of-Distribution objects in this space.

In our hypothetical example (Fig 1), the Out-of-Distribution class (Giant Panda) is represented as a combination of In-Distribution classes (Bear, Zebra, Bird) where 60% of the features of Bear (like body-shape) and 39% of the features of Zebra (like stripes pattern) is 'associated' with the Giant Panda. This is analogous to how children associate unknown objects (Giant Panda) as a combination of recognised objects (Bear and Zebra) when they are asked to describe the unknown object with their learned knowledge [48, 49]. Further, a study by [50] shows that humans can combine available perceptual information with stored

knowledge of experiential regularities, which helps us to describe things that are similar as close and things that are dissimilar as far apart.

Thus, intuitively, all the images of the class Giant Panda should have similar Amalgam Proportion as the hypothetical classifier can associate Giant Panda with some features of Zebra and Bear classes and all the images belong to a single class. Metrics are then computed over the Amalgam Proportion of the Out-of-Distribution class to assess transferability-of-features, (Fig 2). These metrics are based on a different hypothesis of what defines a feature or a class. In the same way, as there are various aspects of robustness, *there are also different interpretations of transferability-of-features*. Therefore, *our metrics are complementary, each highlighting a different perspective of the whole*. The following subsections define them.

### 3.1 Davies–Bouldin Metric (DBM)—Clustering hypothesis

We can use cluster validation techniques to assess the Amalgam Proportion, considering that the cluster of Amalgam Proportion of an Out-of-Distribution class would constitute a class in itself. Here, we choose for simplicity Davies-Bouldin Index [51], one of the most used metrics in internal cluster validation. Davies–Bouldin Metric (DBM) for an Out-of-Distribution class can be defined as follows:

$$\text{DBM} = \left( \frac{1}{n} \sum_{j=1}^{n} |z_j - G|^2 \right)^{1/2}$$

in which, $n$ is the number of samples from the Out-of-Distribution class, $G$ is the centroid of the cluster formed by the soft-labels of all the $n$ samples, and $z$ is soft-label of a single sample of Out-of-Distribution class. A denser cluster would have a lower Davies–Bouldin Metric (DBM) Score representing a consistent view taken by the classifier in terms of features learned from the In-Distribution classes.

### 3.2 Amalgam Metric (AM)—Amalgam hypothesis

Differently from the previous metric, we establish our metric on the hypothesis that the classes learned by a classifier share some similarities with the Out-Of-Distribution class. The classifier can associate this similarity in its features while evaluating these Out-Of-Distribution classes. This hypothesis formulates from the fact that humans can combine available perceptual information with stored knowledge of experiential regularities, which helps us to describe things that are 'similar' as close and things that are 'dissimilar' as far apart [50]. However, what would constitute the baseline Amalgam Proportion for a given Out-of-Distribution class still needs to be determined to assess the extent of the classifier to exploit this existence of similarity between classes.

To calculate the baseline Amalgam Proportion of a given Out-of-Distribution class, we use here the assumption that 'Standard Classifiers' should output a good approximation of the Amalgam Proportion since it is an In-Distribution class for the standard classifier. We thus associate the evaluated Amalgam Proportion of the Raw Zero-Shot Classifier and the baseline Amalgam Proportion of the Standard Classifier for a given class with our Amalgam Metric (AM) (Fig 2) as,

$$\text{AM} = \frac{\|H' - H\|_1}{N - 1} \quad \text{where} \quad H = \sum_{j=1}^{n} z_j, \quad H' = \sum_{j=1}^{n} z'_j$$

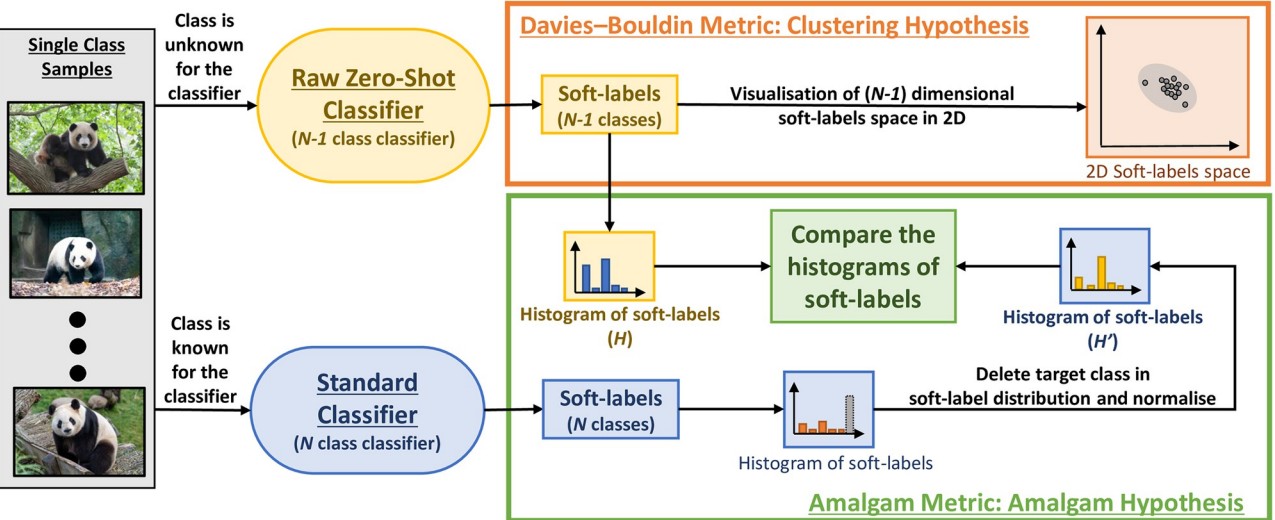

**Fig 2. Illustration of proposed metrics, a) Davies–Bouldin Metric (DBM), and b) Amalgam Metric (AM).** Here, we use a neural network's last layer (classification layer) to evaluate the transferability-of-features from known classes to an unknown class. A network with good transferability-of-features would have a consistent view of the unknown class and form a dense cluster.

in which, $z'$ is the normalized soft-labels of non-target classes from the Standard classifier, and $z$ is the soft-labels of In-Distribution classes of the Raw Zer-Shot Classifier. Note that the given class is 'known' (target) by the standard classifier and is 'unknown' to the Raw Zero-Shot Classifier.

Hence, the Amalgam Metric captures the existence of some unique features learned which are specific to a class which in turn changes the Amalgam Proportion between Raw Zero-Shot Classifier and Standard Classifier. A higher AM score corresponds to a classifier preferring to learn special features of a class over general features present across the distribution. In other words, a lower AM score corresponds to a classifier preferring to learn general features over special features. A non-zero AM score thus verifies the unique special features to a class learned by training the classifier on that specific class.

## 4 Experimental design

### 4.1 Considered datasets

We conduct experiments on three diverse datasets to evaluate the transferability-of-features for the neural networks. We used Fashion MNIST (F-MNIST) [52], CIFAR-10 [53] and a customised Imagenet (Sub) dataset (combining 100 classes of original ILSVRC2012 Imagenet dataset [54] into 10 superclasses) for our evaluations. More details about the customised Sub-Imagenet dataset is mentioned in S1 Appendix in S1 File. Note that the number of samples (7000 for Fashion MNIST, 6000 for CIFAR-10, and roughly 13500 samples for Sub-Imagenet dataset) in the assumed unknown class differ from the dataset. We use the samples from both the training and the testing splits for the 'unknown' class for evaluation because we exclude these samples in the training process.

### 4.2 Considered classifiers

We evaluate different architectures for different datasets. For the Fashion MNIST datasets, we chose to evaluate Multi-Layer Perceptron (MLP), and a shallow Convolution Neural Network

(ConvNet). For the CIFAR-10 dataset, LeNet (a simpler architecture which is a historical mark) [55], VGG (a previous state-of-the-art architecture which is a historical mark) [56], All Convolutional Network (AllConv) (an architecture without max pooling and fully-connected layers) [57], Network in Network (NIN) (an architecture which uses micro neural networks instead of linear filters) [58], Residual Networks (ResNet) (an architecture based on skip connections) [59], Wide Residual Networks (WideResNet) (an architecture which also expands in width) [60], DenseNet (an architecture which is a logical extension of ResNet) [61], and, Capsule Networks (CapsNet) (a completely different architecture based on dynamic routing and capsules) [62]. For our Sub-Imagenet dataset, we chose InceptionV3 [63], and ResNet-50 [59]. More details about the Standard and Raw Zero-Shot Classifiers are mentioned in S2 Appendix in S1 File.

## 4.3 Considered adversarial defences

We also evaluated the transferability-of-features for some of the adversarial defences. Please refer to [23] for a discussion about the performance of adversarial defences in general. for the CIFAR-10 dataset, such as Feature Squeezing (FS) [17], Spatial Smoothing (SS) [17], Label Smoothing (LS) [13], Thermometer Encoding (TE) [20], and Adversarial Training (AT) [18]. We also evaluate classifiers trained with an augmented dataset having Gaussian Noise of $\sigma = 1.0$ (G Aug). More details about the adversarial defences are mentioned in S3 Appendix in S1 File.

## 4.4 Considered attacks

We also evaluated all our standard vanilla classifiers against well-known adversarial attacks such as Fast Gradient Method (FGM) [9], Basic Iterative Method (BIM) [7], Projected Gradient Descent Method (PGD) [18], DeepFool (DF) [64], and NewtonFool (NF) [65]. More details about the adversarial attacks are mentioned in S4 Appendix in S1 File.

## 4.5 Technical assumption and limitations

An assumption is taken that architecture will have a consistent view of all the Out-of-Distribution classes since they belong to the same dataset from which the In-Distribution classes are sampled. An analysis of transferability-of-features for an unknown class belonging to a different dataset is left for future work. Further, in this article, we evaluate a single excluded class as Out-of-Distribution (OoD) class with respect to several In-Distribution classes. Complex analysis on multiple excluded classes and effect on transferability-of-features due to any relationship between excluded (OoD) class and In-Distribution classes is left for future work.

## 5 Experimental results for vanilla classifiers

Table 1 shows the results of our metrics (DBM and AM) for vanilla classifiers. Note that we use mean across all the metric values for $N$ classes of the dataset to be characteristic metric values for an architecture. To enable the visualisation of DBM, we plot a projection of all the points in the decision space of unknown class ($N - 1$ dimensions) into a two-dimensional space. (Fig 3). The characteristic of Isomap is that it seeks a lower-dimensional embedding that maintains geodesic distances between all sample points; that is, it preserves the high-dimensional distance between the points. Other manifold visualisations are added in the S5 Appendix in S1 File.

Similarly, we can also visualise AM, in the form of histograms of soft labels for the classifiers. The computed histograms ($H'$ and $H$) is plotted for every class and classifier to enable the

**Table 1. Mean and standard deviation of Davies–Bouldin Metric (DBM) and Amalgam Metric (AM) scores for vanilla Raw Zero-Shot Classifiers.**

| For Fashion MNIST Dataset | | |
|---|---|---|
| MLP | 0.51±0.09 | 670.71±81.79 |
| ConvNet | 0.47±0.10 | 683.55±76.39 |
| **For Sub-Imagenet Dataset** | | |
| InceptionV3 | 0.56±0.07 | 1335.65±31.83 |
| ResNet-50 | 0.55±0.15 | 1311.97±37.59 |
| **For CIFAR-10 Dataset** | | |
| LeNet | 0.54±0.04 | 115.97±36.92 |
| VGG | 0.61±0.12 | 270.76±186.04 |
| AllConv | 0.64±0.08 | 150.35±39.16 |
| NIN | 0.63±0.09 | 186.14±97.41 |
| ResNet | 0.64±0.13 | 233.84±109.08 |
| DenseNet | 0.61±0.14 | 314.93±130.50 |
| WideResNet | 0.58±0.15 | 417.37±180.78 |
| CapsNet | 0.43±0.03 | 96.96±38.59 |

visualisation of the Amalgam Metric (Fig 4). It is interesting to note that the histograms of CapsNet (Fig 4) are different from the other ones. This reveals that this metric can capture such representation differences. It can be noted (Fig 4) that for most classes of CapsNet, the variation is relatively low than the other architectures. This contributes to having a good representation of CapsNet.

Table 1 reveals that for CIFAR-10 dataset, CapsNet possesses the best transferability-of-features amongst all classifiers examined as it has the least (best) score in both of our metrics. At the same time, LeNet has the second-best transferability. Moreover, other architectures possess similar transferability.

Also for the Sub-Imagenet dataset, both architectures (InceptionV3 and ResNet-50) are equally clustered and predict the Amalgam Proportion similarly. However, ResNet-50 has marginally better transferability than the InceptionV3 as it has better scores for both of our metrics. Similarly, for the Fashion MNIST dataset, both architectures (MLP and ConvNet) have a similar quality of transferability. While ConvNet seems marginally superior to the MLP in terms of clustering the unknown classes more tightly (suggested by DBM), MLP seems marginally superior to predict the Amalgam Proportion better than the ConvNet (suggested by AM).

A further study can also be carried out to analyse the characteristics of representation of the neural network, which makes a class more robust than the other classes. Further investigations can also be carried out to analyse the effect of a class for an adversarial attack based on this. This can also provide insight into the classes which are robust to adversarial attacks. However, these analyses are beyond the scope of the current article and hence, left for future work.

## 6 Link between transferablility of features and adversarial defences

Table 2 shows the results of our metrics (DBM and AM) for vanilla classifiers and classifiers employed with a variety of adversarial defences for improving the robustness of vanilla classifiers for CIFAR-10. We also analyse the statistical relevance of the change in metric values due to introduction of adversarial defences. A paired samples t-test was conducted for our metrics' distributions (DBM and AM) of Vanilla Classifiers (without adversarial defence), and

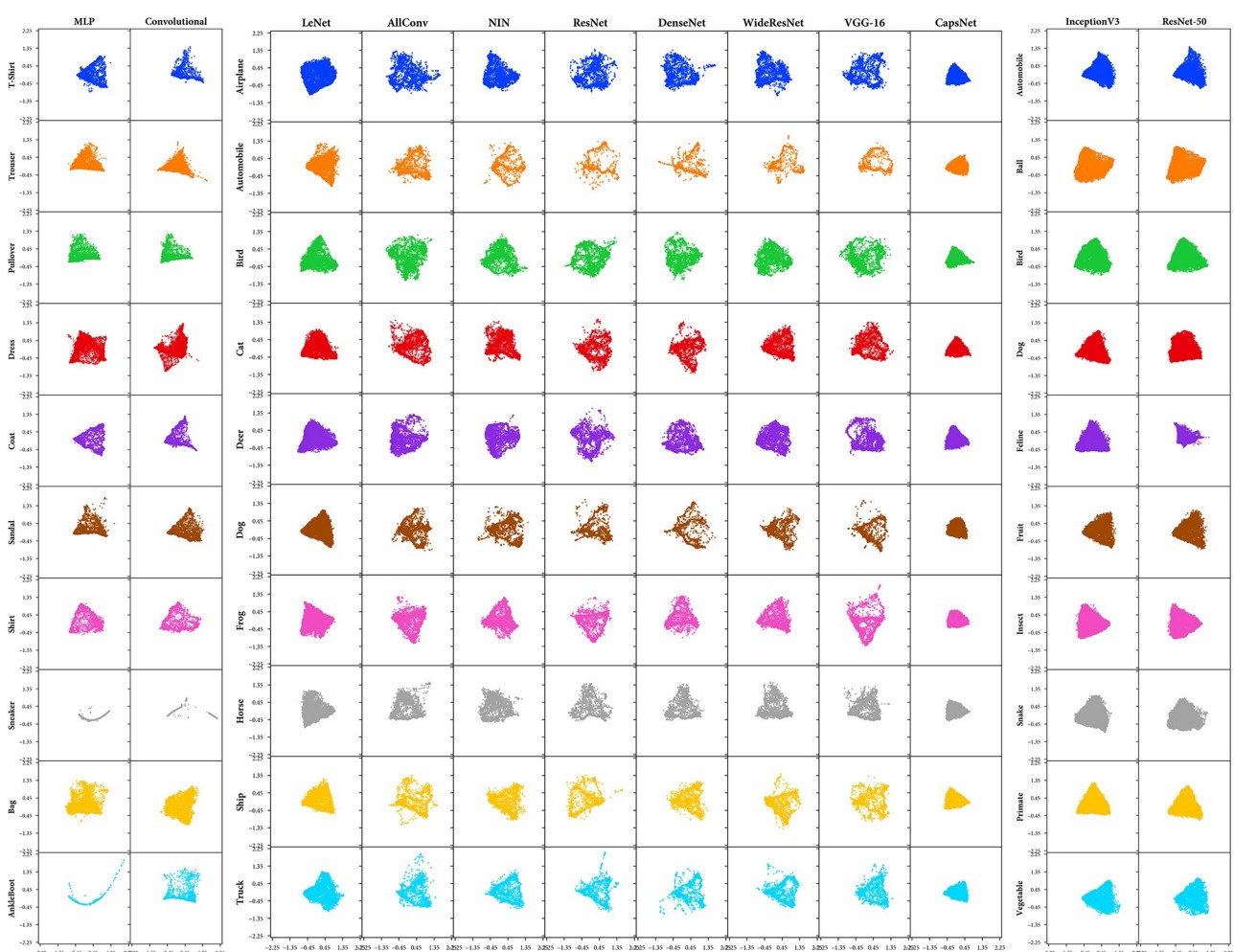

**Fig 3. Visualisation of the Davies–Bouldin Metric (DBM) results for vanilla classifiers using a topology preserving two-dimensional projection with Isometric Mapping (Isomap).** Each row represents a classifier trained with a label excluded whose projection is visualised. While each column represents a different classifier architecture evaluated. A denser clustering represents a consistent view of the unknown class by the architecture.

Adversarially defended classifiers (Table 2) to test the significance in the change in metric values due to Adversarial Defences [66] The Null hypothesis of paired samples t-test assumes that the true mean difference between the distributions is equal to zero. Based on the results (Table 2) adversarial defences, 'in general', tend to improve the transferability-of-features for the neural networks evaluated using Amalgam Proportion. It does so by either by creating a more dense cluster of the soft-labels (suggested by DBM) or learning more general/special features (suggested by AM), or both.

Raw DBM Score values for weaker defences such as Gaussian Augmentation, Feature Squeezing, Spatial Smoothing and Thermometer Encoding lie within the standard deviation of vanilla classifiers suggesting that they affect minimally in clustering the Amalgam Proportion of Out-of-Distribution classes. At the same time, DBM Score values for defences such as LS and AT are noticeably lower than vanilla classifiers suggesting they try to form a denser cluster of Amalgam Proportion compared to the vanilla classifiers. Thus, a better association of

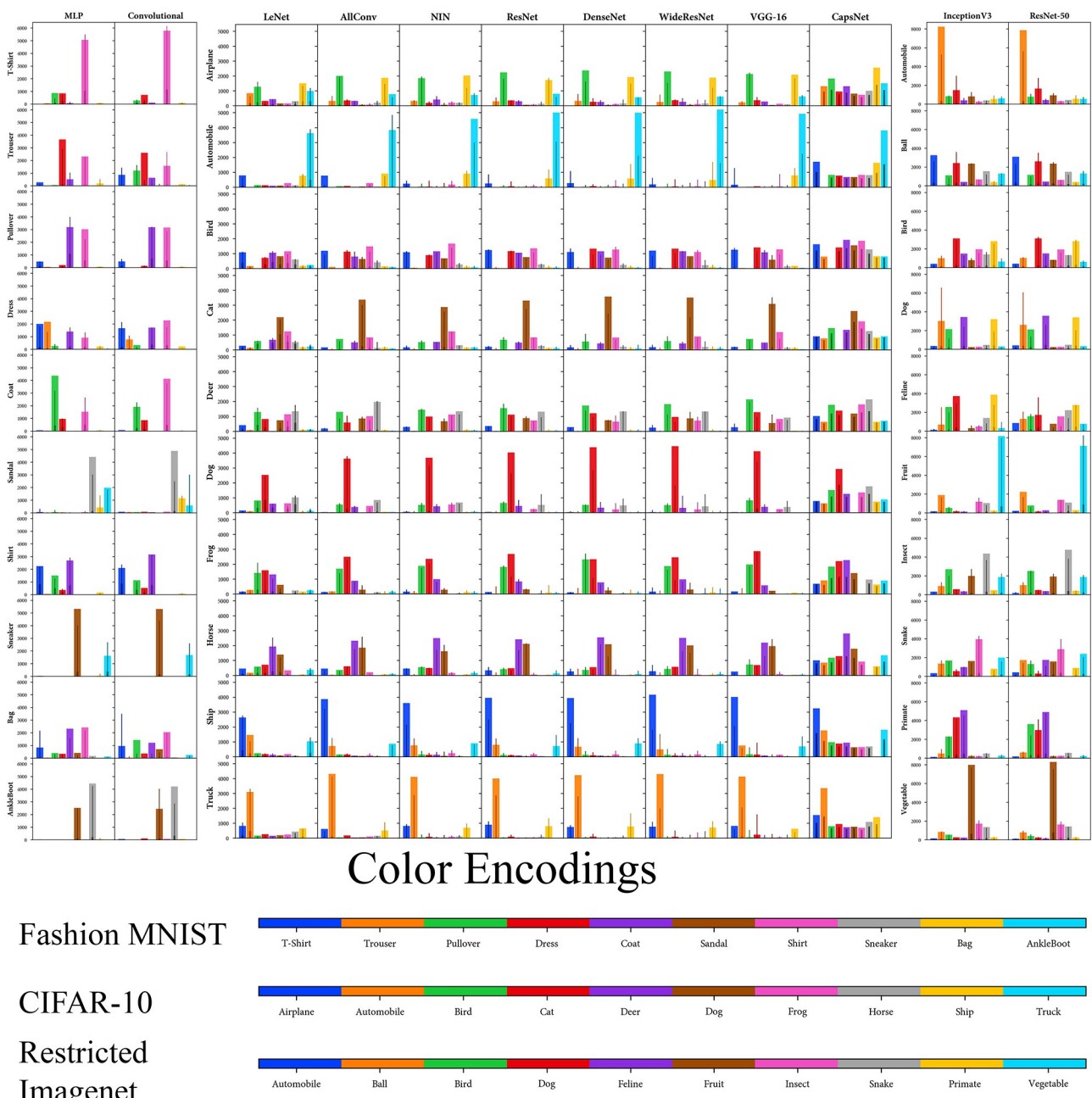

**Fig 4. Histograms of soft-labels ($H'$ and $H$) from which the AM is calculated.** Each row shows the histograms of one classifier with one class excluded. Dark-shaded thinner and light-shaded broader bins are respectively the soft-labels from the ground-truth ($H'$) from the classifier trained on all classes and the soft-labels of the classifier trained on $N - 1$ classes ($H'$).

available features is observed for the more robust defences. From the perspective of AM Score values, the results suggest that LS favours learning special features belonging to a class while AT favours to learn more general features. Interestingly, a general low p-value for the paired samples t-test is observed for the adversarial defences, which suggests that underlying transferability for adversarial defences differ from the vanilla classifiers with high statistical relevance.

**Table 2. Mean and standard deviation of Davies–Bouldin Metric (DBM) and Amalgam Metric (AM) values for different Raw Zero-Shot Classifiers with and without the adversarial defences on CIFAR-10.**

| Davies–Bouldin Metric (DBM) | | | | | | | |
|---|---|---|---|---|---|---|---|
| Architecture | No Defence | Gaussian Augmentation | | Label Smoothing | | Adversarial Training | |
| LeNet | 0.53±0.03 | 0.56±0.04 | (0.00) | 0.43±0.02 | (0.00) | 0.32±0.04 | (0.00) |
| AllConv | 0.61±0.11 | 0.63±0.12 | (0.07) | 0.55±0.10 | (0.00) | 0.47±0.07 | (0.00) |
| Net-in-Net | 0.62±0.11 | 0.66±0.11 | (0.27) | 0.48±0.05 | (0.00) | 0.50±0.06 | (0.00) |
| ResNet | 0.63±0.13 | 0.64±0.11 | (0.17) | 0.52±0.08 | (0.00) | 0.43±0.06 | (0.00) |
| DenseNet | 0.60±0.15 | 0.63±0.14 | (0.09) | 0.54±0.11 | (0.00) | 0.43±0.07 | (0.00) |
| WideResNet | 0.58±0.15 | 0.60±0.15 | (0.05) | 0.55±0.13 | (0.00) | 0.50±0.10 | (0.02) |
| VGG | 0.63±0.13 | 0.59±0.15 | (0.58) | 0.46±0.09 | (0.00) | 0.61±0.10 | (0.13) |
| CapsNet | 0.10±0.01 | 0.23±0.01 | (0.00) | 0.18±0.01 | (0.00) | 0.15±0.02 | (0.00) |
| Architecture | No Defence | Feature Squeezing | | Spatial Smoothing | | Thermometer Encoding | |
| LeNet | 0.53±0.03 | 0.54±0.04 | (0.38) | 0.50±0.03 | (0.01) | 0.52±0.04 | (0.09) |
| AllConv | 0.61±0.11 | 0.62±0.11 | (0.14) | 0.63±0.09 | (0.52) | 0.65±0.05 | (0.27) |
| Net-in-Net | 0.62±0.11 | 0.64±0.08 | (0.20) | 0.63±0.08 | (0.66) | 0.67±0.05 | (0.12) |
| ResNet | 0.63±0.13 | 0.63±0.09 | (0.13) | 0.65±0.06 | (0.39) | 0.65±0.06 | (0.14) |
| DenseNet | 0.60±0.15 | 0.65±0.13 | (0.20) | 0.66±0.11 | (0.61) | 0.71±0.06 | (0.02) |
| WideResNet | 0.58±0.15 | 0.62±0.12 | (0.16) | 0.64±0.11 | (0.57) | 0.69±0.09 | (0.00) |
| VGG | 0.63±0.13 | 0.59±0.14 | (0.13) | 0.62±0.11 | (0.51) | 0.66±0.08 | (0.02) |
| CapsNet | 0.10±0.01 | 0.22±0.01 | (0.00) | 0.21±0.01 | (0.09) | 0.20±0.02 | (0.03) |
| Amalgam Metric (AM) | | | | | | | |
| Architecture | No Defence | Gaussian Augmentation | | Label Smoothing | | Adversarial Training | |
| LeNet | 115.97±36.92 | 84.00±26.39 | (0.03) | 177.08±97.77 | (0.10) | 29.93±16.06 | (0.00) |
| VGG | 270.76±186.04 | 287.75±122.58 | (0.75) | 579.05±121.89 | (0.00) | 218.47±100.50 | (0.44) |
| AllConv | 150.35±39.16 | 153.73±65.96 | (0.90) | 395.28±143.78 | (0.00) | 188.66±67.98 | (0.16) |
| NIN | 186.14±97.41 | 222.68±104.12 | (0.03) | 503.32±145.15 | (0.00) | 86.45±17.60 | (0.01) |
| ResNet | 233.84±109.08 | 266.61±124.12 | (0.17) | 592.57±119.06 | (0.00) | 86.71±46.24 | (0.00) |
| DenseNet | 314.93±130.50 | 303.04±120.54 | (0.70) | 629.48±131.86 | (0.00) | 187.34±71.01 | (0.04) |
| WideResNet | 417.37±180.78 | 443.95±157.46 | (0.13) | 586.84±132.92 | (0.00) | 365.29±199.90 | (0.13) |
| CapsNet | 96.96±38.59 | 111.46±56.69 | (0.07) | 100.01±42.72 | (0.54) | 54.48±20.38 | (0.00) |
| Architecture | No Defence | Feature Squeezing | | Spatial Smoothing | | Thermometer Encoding | |
| LeNet | 115.97±36.92 | 116.85±37.42 | (0.37) | 72.13±20.02 | (0.00) | 272.03±80.86 | (0.00) |
| VGG | 270.76±186.04 | 271.42±184.10 | (0.78) | 183.06±128.16 | (0.02) | 510.39±85.82 | (0.00) |
| AllConv | 150.35±39.16 | 149.47±38.17 | (0.50) | 179.44±68.03 | (0.14) | 537.48±74.51 | (0.00) |
| NIN | 186.14±97.41 | 185.82±100.53 | (0.92) | 148.72±100.69 | (0.00) | 516.72±92.20 | (0.00) |
| ResNet | 233.84±109.08 | 226.19±105.21 | (0.06) | 199.64±99.87 | (0.14) | 531.54±80.03 | (0.00) |
| DenseNet | 314.93±130.50 | 319.33±136.19 | (0.68) | 246.08±99.05 | (0.09) | 585.38±56.48 | (0.00) |
| WideResNet | 417.37±180.78 | 402.62±185.48 | (0.04) | 207.62±131.18 | (0.00) | 646.85±10.66 | (0.00) |
| CapsNet | 96.96±38.59 | 96.95±38.57 | (0.82) | 84.02±31.37 | (0.03) | 280.39±58.42 | (0.00) |

Values in the parentheses are p-values of the paired samples t-test between the metric values of defences and those without defences. A lower DBM score means a more consistent view of the architecture for the unknown class.

## 7 Link between transferablity of features and adversarial attacks

As the results in Table 2, suggests a link between the transferability-of-features and the adversarial defences. It is intuitive to assume that there also exists a link between the transferability-of-features and the adversarial attacks. We conducted a Pearson correlation coefficient test of our metrics (DBM and AM) of the vanilla classifiers with adversarial attacks to evaluate the

**Table 3. Pearson correlation coefficient values of Davies–Bouldin Metric (DBM) and Amalgam Metric (AM) with Mean $L_2$ Score of adversarial attacks for each vanilla classifier and attack pair.**

| Architecture | FGM | BIM | PGD | DF | NF |
|---|---|---|---|---|---|
| | | | DBM with Mean $L_2$ Score | | |
| | | | Fashion MNIST | | |
| MLP | -0.20 (0.58) | -0.17 (0.64) | -0.17 (0.64) | -0.04 (0.91) | -0.02 (0.97) |
| ConvNet | -0.24 (0.50) | -0.30 (0.40) | -0.30 (0.40) | -0.26 (0.46) | -0.22 (0.55) |
| | | | CIFAR-10 | | |
| LeNet | -0.18 (0.61) | -0.70 (0.02) | -0.66 (0.04) | -0.51 (0.13) | -0.36 (0.31) |
| VGG | -0.62 (0.06) | -0.21 (0.55) | -0.20 (0.58) | -0.52 (0.13) | -0.63 (0.05) |
| AllConv | -0.31 (0.39) | -0.56 (0.09) | -0.54 (0.11) | -0.10 (0.78) | -0.30 (0.41) |
| NIN | -0.56 (0.09) | -0.57 (0.08) | -0.57 (0.09) | -0.42 (0.22) | -0.43 (0.21) |
| ResNet | -0.52 (0.12) | -0.76 (0.01) | -0.76 (0.01) | -0.47 (0.17) | -0.51 (0.13) |
| DenseNet | -0.62 (0.06) | -0.50 (0.14) | -0.49 (0.15) | -0.16 (0.65) | -0.22 (0.55) |
| WideResNet | -0.68 (0.03) | -0.75 (0.01) | -0.75 (0.01) | -0.68 (0.03) | -0.75 (0.01) |
| CapsNet | -0.71 (0.02) | -0.45 (0.19) | -0.49 (0.15) | -0.39 (0.26) | -0.48 (0.17) |
| | | | Sub-Imagenet | | |
| InceptionV3 | -0.76 (0.01) | -0.52 (0.13) | -0.52 (0.13) | -0.35 (0.32) | -0.50 (0.14) |
| ResNet-50 | -0.34 (0.34) | -0.12 (0.74) | -0.12 (0.74) | -0.54 (0.10) | -0.25 (0.48) |
| | | | AM with Mean $L_2$ Score | | |
| | | | Fashion MNIST | | |
| MLP | 0.82 (0.00) | 0.26 (0.47) | 0.26 (0.47) | 0.83 (0.00) | 0.84 (0.00) |
| ConvNet | 0.83 (0.00) | -0.07 (0.84) | -0.09 (0.80) | 0.81 (0.00) | 0.82 (0.00) |
| | | | CIFAR-10 | | |
| LeNet | 0.93 (0.00) | 0.32 (0.36) | 0.25 (0.49) | 0.81 (0.00) | 0.89 (0.00) |
| VGG | 0.71 (0.02) | -0.04 (0.91) | -0.07 (0.85) | 0.87 (0.00) | 0.74 (0.01) |
| AllConv | 0.67 (0.03) | 0.42 (0.23) | 0.41 (0.24) | 0.94 (0.00) | 0.73 (0.02) |
| NIN | 0.78 (0.01) | 0.84 (0.00) | 0.84 (0.00) | 0.96 (0.00) | 0.89 (0.00) |
| ResNet | 0.35 (0.32) | 0.57 (0.09) | 0.57 (0.09) | 0.79 (0.01) | 0.83 (0.00) |
| DenseNet | 0.53 (0.11) | 0.78 (0.01) | 0.78 (0.01) | 0.78 (0.01) | 0.84 (0.00) |
| WideResNet | 0.66 (0.04) | 0.68 (0.03) | 0.68 (0.03) | 0.78 (0.01) | 0.68 (0.03) |
| CapsNet | 0.98 (0.00) | 0.69 (0.03) | 0.73 (0.02) | -0.17 (0.63) | 0.47 (0.17) |
| | | | Sub-Imagenet | | |
| InceptionV3 | 0.75 (0.01) | 0.14 (0.70) | 0.14 (0.70) | 0.28 (0.44) | 0.25 (0.49) |
| ResNet-50 | 0.82 (0.00) | 0.31 (0.39) | 0.31 (0.39) | 0.51 (0.13) | 0.50 (0.15) |

Values in the parentheses are p-values of the Pearson correlation test. Higher absolute Pearson correlation values signify a stronger relationship between the metrics, and a lower p-value signifies the higher confidence in the correlation values.

statistical relevance of this link between transferability evaluated using Amalgam Proportion, and adversarial attacks [67] The Pearson correlation analysis of our metrics suggests a relationship between our metrics and the adversarial attacks in general.

We use the analysis of adversarial attacks in the form of Mean $L_2$ Score ($L_2$ difference between the original sample and the adversarial one) to compute the correlation [64]. The Pearson correlation coefficients of our metrics (DBM and AM) with Mean $L_2$ Score is shown in Table 3 for every architecture and attack. Moreover, these Pearson relationships between our metrics and Mean $L_2$ Score can also be visualised (Figs 5 and 6). We visualise the Pearson correlation between the Raw Zero-Shot metrics (DBM and AM) with the adversarial metrics

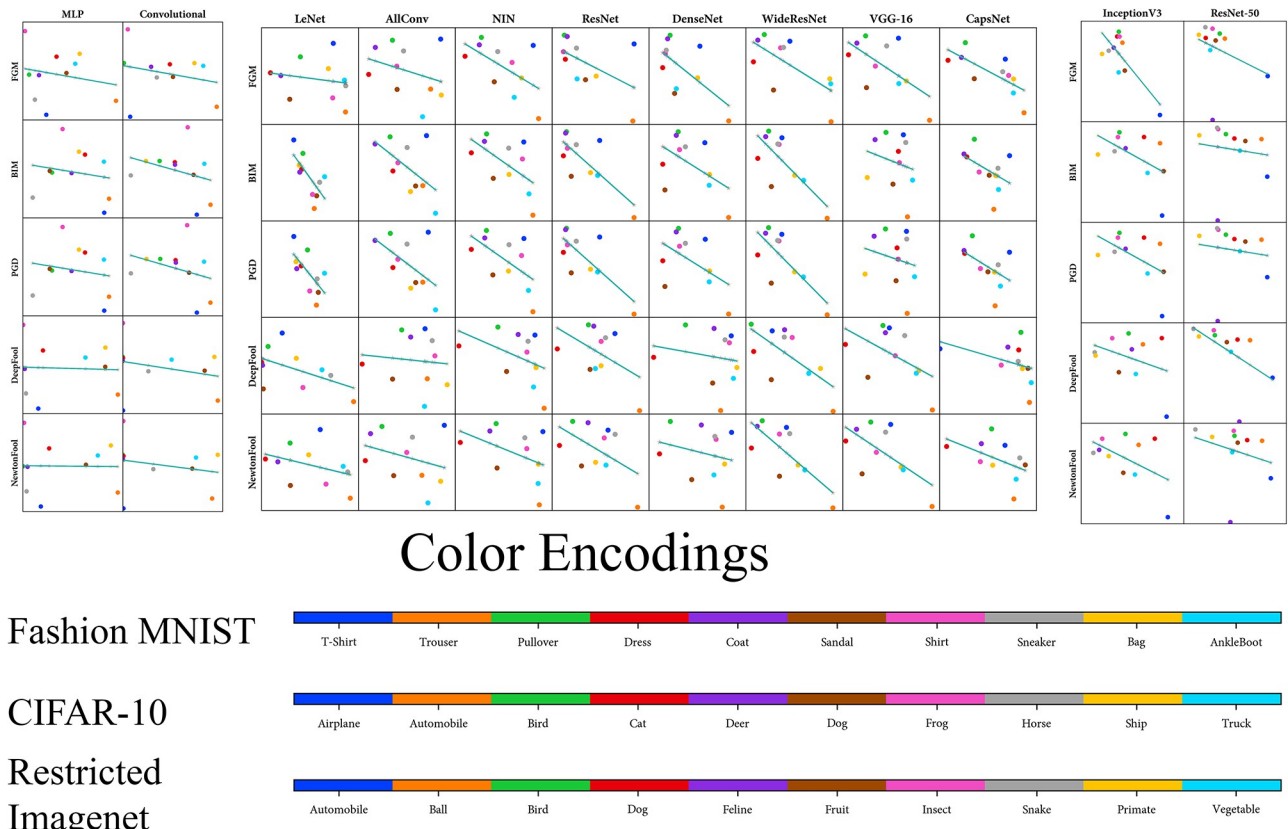

**Fig 5. Visualisation of Pearson correlation of Davies-Bouldin Metric (DBM) with Mean $L_2$ Score of adversarial attacks (Table 3).** Here, the x-axis represents the Mean $L_2$ Scores while the y-axis represents the DBM values. Each point represent a DBM value and Mean $L_2$ Score for a labelled class. Slope of the line determines the pearson correlation coefficient value while lower dispersion of points from the line determines a low p-Value.

(Adversarial Accuracy and Mean $L_2$ Score) mentioned in Table 3. Figs 5 and 6, visualizes the relationship of Raw Zero-Shot metrics with adversarial metrics.

## 8 General discussion on transferability-of-features

On carefully observing the metric values (Tables 1–3), we found that our assessment of representation quality using Amalgam Proportion also explains some of the propositions by other researchers, we highlight some of our key findings below,

### Does a model with high capacity will have a better transferability-of-features?

Our results reveal that a deeper network which generally has a higher capacity [18] does not necessarily correspond to have a better transferability-of-features. As CapsNet and LeNet, which are much shallower than the other deeper networks, are shown to have superior transferability than other deeper networks (Table 2).

### Why CapsNet has better transferability than other deeper networks?

We observe that Capsule Networks (CapsNet) has the best transferability-of-features amongst other neural networks (Table 2). Our results suggest that CapsNet produces a denser cluster

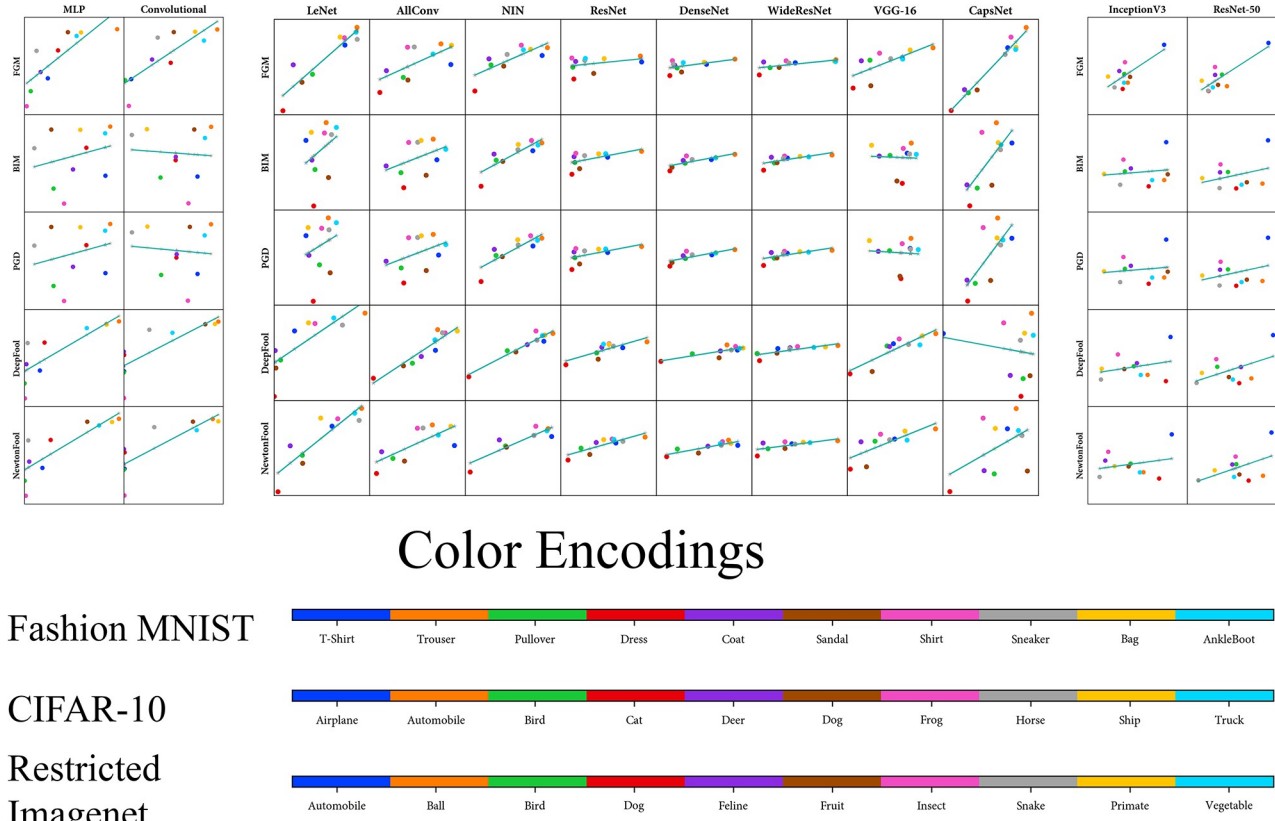

## Color Encodings

**Fashion MNIST**

| T-Shirt | Trouser | Pullover | Dress | Coat | Sandal | Shirt | Sneaker | Bag | AnkleBoot |

**CIFAR-10**

| Airplane | Automobile | Bird | Cat | Deer | Dog | Frog | Horse | Ship | Truck |

**Restricted Imagenet**

| Automobile | Ball | Bird | Dog | Feline | Fruit | Insect | Snake | Primate | Vegetable |

**Fig 6. Visualisation of Pearson correlation of Amalgam Metric (AM) with Mean $L_2$ Score of adversarial attacks (Table 3).** Here, the x-axis represents the Mean $L_2$ Scores while the y-axis represents the AM values. Each point represent an AM value and Mean $L_2$ Score for a labelled class.

for Amalgam Proportion and learns more general features. We believe it might be because of the dynamical nature (routing) of the CapsNet. Thus, our results call for a more in-depth investigation of Capsule Networks and their property of transferring features.

### How does augmenting the dataset with Gaussian Noise affect the transferability-of-features?

We observe that Gaussian Augmentation degrades the transferability-of-features of all the classifiers (Table 2). This supports our intuition (Section 3), as adding Gaussian noise to the images subdue the features of the image by blurring, making the classifier harder to interpret these features. Consequently, a weaker association of the transferability with these features is observed through the perspective of Amalgam Proportion.

### How does Label Smoothing improve the transferability-of-features?

Our results corroborate the analysis in [68] that Label Smoothing (LS) encourages the features to group in tight, equally distant clusters. The raw metric values from our experiments for LS suggests that classifiers employed with LS do form a tighter cluster in soft-label space (as suggested by DBM) (Table 2). At the same time, LS also favours the classifiers to learn special features belonging to a class (as suggested by AM).

### Does learning features near to the feature centroid is beneficial against adversarial attacks?

It is shown that forcing a loss function to make features near to the feature centroid is beneficial against adversarial attacks in [69]. We notice from Table 3 that Davies–Bouldin Metric (DBM) is related quite reasonably with the adversarial attacks. Our metric DBM, as it calculates precisely the closeness of the feature to the feature centroid, and is related quite reasonably with the adversarial attacks corroborates the results by [69].

## 9 On links of transferability-of-features with adversarial attacks and defences

Based on our experiments and results, we hypothesise that the cause of the links of transferability-of-features is due to the presence of a bias introduced in the training of neural networks. We call this bias—Dataset Bias and define it as a bias towards the classes and data distribution present in a dataset. It is already proven theoretically that it is possible to separate any number of classes, provided enough samples are evaluated. However, this separation only exists inside the evaluated samples' underlying data distribution and classes. With the introduction of noise or corruptions in the underlying data distribution, this separation of classes is not valid anymore as the distribution is substantially modified. The area related to Zero-Shot Learning and Transfer Learning investigates this bias by introducing unknown class samples when inferring. In contrast, in adversarial machine learning, the same bias is studied by introducing noisy adversarial samples.

## 10 Conclusions

This article proposes a novel Zero-Shot learning-based method, entitled Raw Zero-Shot, to assess the transferability-of-features in neural networks. In order to assess the transferability, two associated metrics are formally defined based on different hypotheses of interpreting transferability-of-features. Our results suggest that CapsNet, a dynamic routing network, has the best transferability-of-features amongst classifiers which calls for a more in-depth investigation of Capsule Networks. Also, the behaviour of different architectures spotted in the DBM can be visualised in the Isomap plots, which shows that the DBM indeed capture the existing differences in transferability-of-features.

Our experimental results reveal that,

- Classifiers employed with adversarial defences improve the transferability-of-features for the classifiers as evaluated by DBM, suggesting that to improve the robustness of the classifiers, we have to improve the transferability-of-features of the classifiers too.

- Adversarial defences have a low p-value (in general) in the paired samples t-test when compared to vanilla classifiers in general, suggesting that transferability is significantly affected by various adversarial defences.

- A high Pearson correlation coefficient and low p-value (in general) of the Pearson correlation test between DBM and the adversarial attacks suggest a link between the transferability-of-features and the adversarial attacks.

Hence, the proposed Raw Zero-Shot was able to assess and understand the transferability-of-features from the perspective of Out-of-Distribution classes of different neural networks' architectures, along with the adversarial defences and link this property of transferability of the neural networks with adversarial attacks and defences. It also opens up new possibilities of

using transferability-of-features for both evaluation (i.e. as a quality assessment) and the development (e.g. as a loss function) of neural networks.

## Supporting information

**S1 File.**
(PDF)

## Acknowledgments

We would like to thank Prof. Junichi Murata for his kind support without which it would not be possible to conduct this research.

## Author Contributions

**Conceptualization:** Shashank Kotyan, Moe Matsuki, Danilo Vasconcellos Vargas.

**Data curation:** Shashank Kotyan, Moe Matsuki.

**Formal analysis:** Shashank Kotyan, Moe Matsuki.

**Funding acquisition:** Danilo Vasconcellos Vargas.

**Investigation:** Shashank Kotyan, Moe Matsuki.

**Methodology:** Shashank Kotyan, Moe Matsuki.

**Project administration:** Danilo Vasconcellos Vargas.

**Resources:** Moe Matsuki, Danilo Vasconcellos Vargas.

**Software:** Moe Matsuki, Danilo Vasconcellos Vargas.

**Supervision:** Danilo Vasconcellos Vargas.

**Validation:** Shashank Kotyan, Moe Matsuki.

**Visualization:** Shashank Kotyan, Moe Matsuki.

**Writing – original draft:** Shashank Kotyan, Moe Matsuki.

**Writing – review & editing:** Shashank Kotyan, Moe Matsuki, Danilo Vasconcellos Vargas.

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
