## [Decision Letter · Decision Letter 0]

1 Feb 2022

PONE-D-22-00931Transferability Of Features For Neural Networks Links To Adversarial Attacks and DefencesPLOS ONE

Dear Dr. Kotyan,

Thank you for submitting your manuscript to PLOS ONE. After careful consideration, we feel that it has merit but does not fully meet PLOS ONE’s publication criteria as it currently stands. Therefore, we invite you to submit a revised version of the manuscript that addresses the points raised during the review process.

We look forward to receiving your revised manuscript.

Kind regards,

Sathishkumar V E

Academic Editor

PLOS ONE

Journal Requirements:

- https://arxiv.org/abs/1906.06627

In your revision ensure you cite all your sources (including your own works), and quote or rephrase any duplicated text outside the methods section. Further consideration is dependent on these concerns being addressed.

"This work was supported by JST, ACT-I Grant Number JP-50243 and JSPS KAKENHI Grant Number JP20241216. "

"DV, JST, ACT-I Grant Number JP-50243, https://www.jst.go.jp/kisoken/act-i/index.html

DV, JSPS KAKENHI Grant Number JP20241216, " ext-link-type="uri" xlink:type="simple">https://www.jsps.go.jp/english/e-grants/"

4. We note that Figures 1 and 2 in your submission contain copyrighted images. All PLOS content is published under the Creative Commons Attribution License (CC BY 4.0), which means that the manuscript, images, and Supporting Information files will be freely available online, and any third party is permitted to access, download, copy, distribute, and use these materials in any way, even commercially, with proper attribution. For more information, see our copyright guidelines: http://journals.plos.org/plosone/s/licenses-and-copyright.

a. You may seek permission from the original copyright holder of Figures 1 and 2 to publish the content specifically under the CC BY 4.0 license. 

5 Please review your reference list to ensure that it is complete and correct. If you have cited papers that have been retracted, please include the rationale for doing so in the manuscript text, or remove these references and replace them with relevant current references. Any changes to the reference list should be mentioned in the rebuttal letter that accompanies your revised manuscript. If you need to cite a retracted article, indicate the article’s retracted status in the References list and also include a citation and full reference for the retraction notice.

Reviewers' comments:

Reviewer's Responses to Questions

**Comments to the Author**

1. Is the manuscript technically sound, and do the data support the conclusions?

Reviewer #1: Yes

Reviewer #2: Yes

2. Has the statistical analysis been performed appropriately and rigorously? 

Reviewer #1: Yes

Reviewer #2: Yes

3. Have the authors made all data underlying the findings in their manuscript fully available?

Reviewer #1: Yes

Reviewer #2: Yes

4. Is the manuscript presented in an intelligible fashion and written in standard English?

Reviewer #1: Yes

Reviewer #2: Yes

5. Review Comments to the Author

Reviewer #1: This article proposes a novel Zero-Shot learning-based method, entitled Raw Zero-Shot, to assess the transferability-of-features in neural networks. In order to assess the transferability, two associated metrics are formally defined based on different hypotheses of interpreating transferability-of-features. This article is sufficiently novel and the contributions are good. This paper may be considered after addressing the below:

1. The section numbers to be mentioned and it is confuse to identify the sections and subsections.

2. The literature study extended with the recently published articles.

3. Discuss the reasons to achieve the superior performance of the proposed models.

4. Discuss about the remaining sections in the introduction.

5. how about the computational time complexity of the proposed model?

6. Discuss the limitations of the model.

Reviewer #2: 1. Abstract can conclude with qualitative results of the proposed work

2. Literature review section can be included to discuss works done so far related to the problem

3. Overall grammar has to be checked

6. PLOS authors have the option to publish the peer review history of their article (what does this mean?). If published, this will include your full peer review and any attached files.

Reviewer #1: No

Reviewer #2: No

---

## [Author Response · Author response to Decision Letter 0]

28 Feb 2022

Reviewer #1: The section numbers to be mentioned and it is confuse to identify the sections and subsections. Discuss about the remaining sections in the introduction.

Response: The readability of the article was improved by mentioning section numbers and discussing the sections in the introduction according to the feedback. 

Reviewer #1: The literature study extended with the recently published articles.

Response: The literature study is revised to discuss some recently published articles in the related fields.

Reviewer #1: Discuss the reasons to achieve the superior performance of the proposed models.

Response: Despite varying types of adversarial attacks, adversarial defences and neural network architectures, our experiments suggest that models which have better transferability-of-features as measured with our metrics were more robust to adversarial attacks. The transferability-of-features for a few interesting models are discussed in Section 8 of the article. Further, a hypothesis of the effect of dataset bias on the transferability-of-features is presented in Section 9 of the article.

Reviewer #1: How about the computational time complexity of the proposed model?

Response: Our approach follows a leave-one-out strategy to exclude a class keeping all other parameters the same as the standard model. Therefore, the evaluation of a single class has the same computational time complexity as training the model with other classes. Since the model is trained on “n-1” classes instead of “n” classes. However, evaluating all the classes of the dataset requires “n” pieces of training, where “n” is the number of classes in the dataset. 

Reviewer #1: Discuss the limitations of the model.

Response: The technical difficulty and limitations of the proposed scheme are now discussed in the revised Section 4 Experimental Design.

Reviewer #2: Literature review section can be included to discuss works done so far related to the problem. 

Response: The literature study is revised to discuss some recently published articles in the related fields.

Reviewer #2: Abstract can conclude with qualitative results of the proposed work. 

Response: Thank you for pointing this out. To improve our abstract, we have included the qualitative inference from our experiments. 

Reviewer #2: Overall grammar has to be checked

Response: Our article is revised and has been corrected for grammatical and typing mistakes.

---

## [Decision Letter · Decision Letter 1]

14 Mar 2022

Transferability Of Features For Neural Networks Links To Adversarial Attacks and Defences

PONE-D-22-00931R1

Dear Dr. Kotyan,

We’re pleased to inform you that your manuscript has been judged scientifically suitable for publication and will be formally accepted for publication once it meets all outstanding technical requirements.

Kind regards,

Sathishkumar V E

Academic Editor

PLOS ONE

Additional Editor Comments (optional):

Reviewers' comments:

Reviewer's Responses to Questions

**Comments to the Author**

1. If the authors have adequately addressed your comments raised in a previous round of review and you feel that this manuscript is now acceptable for publication, you may indicate that here to bypass the “Comments to the Author” section, enter your conflict of interest statement in the “Confidential to Editor” section, and submit your "Accept" recommendation.

Reviewer #1: All comments have been addressed

2. Is the manuscript technically sound, and do the data support the conclusions?

Reviewer #1: Yes

3. Has the statistical analysis been performed appropriately and rigorously? 

Reviewer #1: Yes

4. Have the authors made all data underlying the findings in their manuscript fully available?

Reviewer #1: Yes

5. Is the manuscript presented in an intelligible fashion and written in standard English?

Reviewer #1: Yes

6. Review Comments to the Author

Reviewer #1: The authors addressed all the comments and this version may be considered for publication in this journal.

7. PLOS authors have the option to publish the peer review history of their article (what does this mean?). If published, this will include your full peer review and any attached files.

Reviewer #1: No

---

## [Editor Report · Acceptance letter]

1 Apr 2022

PONE-D-22-00931R1 

Transferability Of Features For Neural Networks Links To Adversarial Attacks and Defences 

Dear Dr. Kotyan:

I'm pleased to inform you that your manuscript has been deemed suitable for publication in PLOS ONE. Congratulations! Your manuscript is now with our production department. 

Kind regards, 

on behalf of

Dr. Sathishkumar V E 

Academic Editor

PLOS ONE